# Removal of Different Kinds of Heavy Metals by Novel PPG-nZVI Beads and Their Application in Simulated Stormwater Infiltration Facility

**Xiaoran Zhang** [1,2,3] ⬤, **Lei Yan** [1], **Junfeng Liu** [4,*], **Ziyang Zhang** [1,2,3] **and Chaohong Tan** [1]

1 Key Laboratory of Urban Stormwater System and Water Environment, Ministry of Education, Beijing University of Civil Engineering and Architecture, Beijing 102616, China; zhangxiaoran@bucea.edu.cn (X.Z.); yanlei_1992yl@163.com (L.Y.); zhangziyang@bucea.edu.cn (Z.Z.); chaohongtan@126.com (C.T.)
2 Beijing Engineering Research Center of Sustainable Urban Sewage System Construction and Risk Control, Beijing University of Civil Engineering and Architecture, Beijing 100044, China
3 Beijing Advanced Innovation Center for Future Urban Design, Beijing 100044, China
4 Department of Water Conservancy and Civil Engineering, Beijing Vocational College of Agriculture, Beijing 102442, China
* Correspondence: liujunfeng@bvca.edu.cn

**Abstract:** Polyvinyl alcohol and pumice synthetized guar gum-nanoscale zerovalent iron beads (PPG-nZVI beads) were synthesized, and their adsorption towards $Pb^{2+}$, $Cu^{2+}$, and $Zn^{2+}$ ions was evaluated. The adsorption kinetics of metal ions was well fitted by the pseudo-second-order model. The adsorption rate decreased followed in the order of $Cu^{2+} > Pb^{2+} > Zn^{2+}$, consistent with the reduction potential of the ions. The sorption isotherm was well fitted by Langmuir model. The maximum adsorption capacity decreased followed in the order of $Pb^{2+} > Cu^{2+} > Zn^{2+}$, which suggested that the strength of covalent bonds between the metal ions and surface functional groups substituted to the beads is one of the major factors in the adsorption process. Adsorption increased with the increase of pH and the largest sorption occurred at pH 5.5, while ionic strength did not significantly influence the adsorption process. The application of PPG-nZVI beads as filling materials in the simulated stormwater infiltration facility shows good removal efficiency in treating the contaminated water containing $Pb^{2+}$, $Cu^{2+}$, $Zn^{2+}$, $Cr^{6+}$, and $Cd^{2+}$ and the removal rate was more than 65% at least. The results indicated that the PPG-nZVI beads could be applied as promising sorbents for purification of heavy metal contaminated water.

**Keywords:** zerovalent iron; beads; heavy metal; stormwater; infiltration; adsorption

## 1. Introduction

Heavy metals with density more than 5.0 g/cm$^3$ [1] bring significant impacts on the environment and human. $Pb^{2+}$, $Cu^{2+}$, and $Zn^{2+}$ are typical heavy metals that commonly existed in urban runoff and are difficult to be degraded. Heavy metals prefer to accumulate in organisms and much of them are known to be carcinogenic and highly toxic [2]. They have attracted great attention due to their potential hazards to ecosystems and human health. Varieties of methods such as chemical precipitation, reverse osmosis, electrochemical treatment, ion exchange, membrane filtration, and adsorption have been used to remove heavy metals from wastewater [3]. Among them, adsorption is found to be one of the promising techniques with advantages such as high efficiency, environmental friendly, and low operational and maintenance costs.

Nano adsorbents have exhibited obvious advantages due to their large specific surface area and great number of surface active sites. Especially, zero valent iron nanoparticles (nZVI) have

attracted much attention to treat wastewater owing to its small size, amorphous structure and the presence of defective sites. The nZVI could act as promising regents that have been used to treat pollutants such as chlorinated organic solvents, nitrate, heavy metals and azo-dyes [4]. Furthermore, there are multiple interaction mechanisms between nZVI and compounds [5], such as surface mediated reduction, precipitation, surface complexation, partial chemical reduction, surface oxidation, and surface adsorption. Studies show that they are unique in environmental remediation [6].

However, due to the tiny particle size and high surface energy of the nano materials, nZVI are difficult to be separated from the wastewater, which limits their direct applications. A very straightforward and effective way is to prepare the well-structured novel beads in order to be applied as adsorbents. In addition, besides the unique properties of nanomaterials that mixed in the beads, other agents used during the synthesis procedure of the beads could also be verified not only used for forming the beads shape but also helpful on removal of the target pollutants to some extent. Previously, we have successfully prepared the polyvinyl alcohol (PVA) and pumice synthetized guar gum-nZVI beads (PPG-nZVI beads) and found their great removal efficiency towards lead [7]. The PPG-nZVI beads we have synthesized show superior properties such as large surface area, uniform size, high mechanical strength and good chemical stability. Their excellent removal efficiency towards lead also proved that they have the potential to be applied as adsorbents to remove heavy metal ions from wastewater.

In this study, we further examined the removal efficiency of PPG-nZVI beads towards different kind of heavy metal ions as well as the influence of pH and ionic strength on their removal. As one of the typical commonly used technologies, stormwater infiltration facility is efficient to remove pollutants from urban runoff. The facility could be designed in several patterns such as planting pool, permeable pavement system, and bioretention system. The common fillers in the facilities are materials such as sand, zeolite, sawdust, and gravel. To improve the treating efficiency of the filler, we attempt to use the PPG-nZVI beads as one types of filling materials in stormwater infiltration facility to examine the removal efficiency towards heavy metals in runoff. Our objectives are: i) To determine the adsorption kinetics and isotherms, in order to investigate the possible heavy metal removal mechanisms by PPG-nZVI beads and to distinguish the adsorption behavior of different heavy metal ions (i.e., $Pb^{2+}$, $Cu^{2+}$, and $Zn^{2+}$); ii) to examine the effect of different conditions including solution pH and ionic strength on the sorption process in order to check the applicable of the beads in wide environmental conditions; iii) to apply PPG-nZVI beads as filling materials in the simulated stormwater infiltration facility and examine their removal efficiency on the multi-heavy metal polluted runoff including $Pb^{2+}$, $Cu^{2+}$, $Zn^{2+}$, $Cr^{6+}$, and $Cd^{2+}$. This study will provide a new insight to better understanding of the adsorption mechanisms of heavy metals by PPG-nZVI beads as well as the application of the novel beads in remediation area especially for treating polluted runoff.

## 2. Materials and Methods

### 2.1. Materials

Analytical grade reagents including $FeSO_4·7H_2O$, $KBH_4$, anhydrous ethanol, the standard $Pb^{2+}/Cu^{2+}/Zn^{2+}/Cr^{6+}/Cd^{2+}$ solution and PVA-124 were obtained from Sinopharm Chemical Reagent Beijing Co., Ltd., China. Pumice particles were supplied by Jiang Teng ash development co., Ltd., Yunnan, China. Guar gum was purchased from Beijing Guar Science and Trading, Ltd. Synthesis procedures of the PPG-nZVI beads were similar as our previous study [7]. In brief, the guar gum-stabilized nZVI was firstly prepared following the methods reported by Geng et al. [8]. Secondly, 0.10 g the above guar gum-stabilized nZVI powder was mixed with 0.05 g pumice particles. They were then blend with 3.00 g PVA and diluted into 300 mL deoxidation deionized water by stirring to form flocculent precipitate. After reaction for 5 min, the precipitate was then quickly modified into beads with funnels and specialized beads forming boards. Finally, the fresh made PPG-nZVI beads were stored in deoxidation anhydrous ethanol immediately. The ratio of chemicals was optimized in order

to prepare the well-formed and strengthen beads. From the dry-wet alternating test, the beads show good swelling property when they were immersed into water and could still maintained the formation after drying. They could keep the spherical shape and show strong stability. The beads also show high mechanical resistance under certain strength of pressing force without deformation.

## 2.2. Sorption Experiments

The standard solution ($1000 \text{ mg·L}^{-1}$) of $Pb^{2+}$, $Cu^{2+}$, or $Zn^{2+}$ was diluted to the target concentration levels by deionized water as experimental working solutions.

In the sorption kinetic experiments, 2.00 g PPG-nZVI beads were added into 400 mL of $Pb^{2+}$, $Cu^{2+}$ or $Zn^{2+}$ solutions. The supernatant was taken out at different shaking time intervals in order to analyze the concentration of heavy metal. To measure the sorption isotherms of each heavy metal ion by PPG-nZVI beads, 0.25 g PPG-nZVI beads were added to 50 mL of various of concentration ($1-80 \text{ mg·L}^{-1}$) with the heavy metal working solution in a series of 50 mL centrifuge tubes. All tubes were shaken in a temperature-controlled shaker ($150 \text{ r·min}^{-1}$, 25 °C ± 1) for 24 h which was long enough to reach equilibrium as tested by the kinetic study. In kinetic and isotherm study, the solution pH is 5.0.

To study the influence of initial pH on the removal of $Cu^{2+}$, the initial pH of the solutions was adjusted to 3.0, 4.0, 5.0, and 5.5. To examine the effect of ionic strength on sorption, the concentration of $Na^+$ in $20 \text{ mg·L}^{-1}$ $Pb^{2+}$, $Cu^{2+}$ or $Zn^{2+}$ solution was adjusted to 0.00 M, 0.01 M, 0.05 M, 0.10 M, 0.50 M, and 1.00 M with sodium nitrate as background solution. The dosage of PPG-nZVI beads in the solution was $5.00 \text{ g·L}^{-1}$. The initial $Pb^{2+}$, $Cu^{2+}$ or $Zn^{2+}$ concentration was $20.0 \text{ mg·L}^{-1}$. After reaching equilibrium, the beads were separated from the aqueous phase. The concentration of heavy metal in the supernatant was analyzed by ICP-OES 710.

The pseudo first order model and the pseudo second order model are used to fit the data of sorption kinetics. The models are presented using the following Equations (1) and (2):

$$q_t = q_e \left(1 - e^{-k_1 t}\right) \tag{1}$$

$$q_t = \frac{k_2 q_e^2 t}{1 + k_2 q_e t} \tag{2}$$

where $q_t$ ($\text{mg·g}^{-1}$) and $q_e$ ($\text{mg·g}^{-1}$) are the amounts of the metals adsorbed on per unit of adsorbent at any time $t$ (h) and equilibrium, respectively; $k_1$ ($\text{h}^{-1}$) and $k_2$ ($\text{g·mg}^{-1}\text{·h}^{-1}$) are the constants of the kinetic model, respectively. The nonlinear forms of Langmuir model and Freundlich model are adopted to fit the sorption data. The models are expressed as Equations (3) and (4):

$$q_e = q_m K_L \frac{C_e}{1 + K_L C_e} \tag{3}$$

$$q_e = K_F C_e^n \tag{4}$$

where $q_e$ ($\text{mg·g}^{-1}$) is the amount of the adsorbate on the surface of the adsorbent at equilibrium; $C_e$ ($\text{mg·L}^{-1}$) is the equilibrium concentration of the adsorbate in solution; $q_m$ ($\text{mg·g}^{-1}$) is the maximum sorption capacity generated from the Langmuir model; $K_L$ ($\text{L·mg}^{-1}$) is a constant for Langmuir model related to the energy of sorption and the affinity of the binding sites; while $K_F$ (($\text{mg·g}^{-1}$) ($\text{L·mg}^{-1}$)$^n$) is the Freundlich constants related to sorption capacity; n is the empirical parameter varied with the degree of heterogeneity of adsorbing sites. Data analysis was carried out by Origin 8.0.

## 2.3. Characterization

The surface functional groups of PPG-nZVI beads before and after adsorption of $Pb^{2+}$, $Cu^{2+}$ or $Zn^{2+}$ were characterized by Fourier transform infrared spectroscopy (FTIR) (Bruker, Tensor 27).

The surface composition and the element valence of PPG-nZVI beads was characterized by X-ray photoelectron spectroscope (Thermo escalab 250XI).

### 2.4. Application of PPG-nZVI Beads in Simulated Stormwater Infiltration Facility

To examine the removal efficiency of PPG-nZVI beads in remediation system, PPG-nZVI beads (particle size 0.6–1 cm, thickness 3 cm) were paved as one layer in the simulated stormwater infiltration facility (named as pool 1) to treat the heavy metal wastewater. Besides the beads layer, the traditional filling materials including sand (≤2.5 cm, thickness 5 cm), sawdust (thickness 3 cm), pumice (particle size 1.5–2.5 cm, thickness 15 cm) and fine gravel (particle size 1–2 cm, thickness 15 cm) as other layers were set under the beads layer. The pool without beads layer (named as pool 2) was also set up to verify the effect PPG-nZVI beads layer on treating heavy metals. Multi-heavy metal solution with the concentration of 10 mg·L$^{-1}$ for Pb$^{2+}$, 5 mg·L$^{-1}$ for Cu$^{2+}$, 30 mg·L$^{-1}$ for Zn$^{2+}$, 5 mg·L$^{-1}$ for Cr$^{6+}$, and 5 mg·L$^{-1}$ for Cd$^{2+}$ was flowed from the inlet. The devices run in a continuous feed mode and were set to 100 L·h$^{-1}$ by the constant current pump for 3 h. Water samples in the outlet were taken out every 15 min and analyzed. The infiltration pool and water intake diagram were shown in Figure 1. The length, width and height of the devices were all settled to be 50 cm.

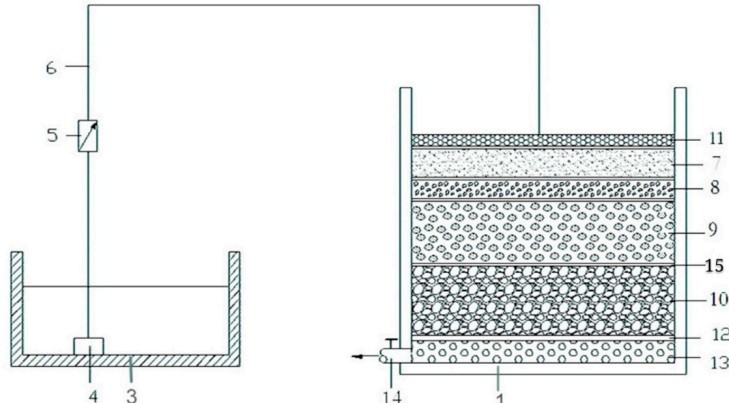

**Figure 1.** Infiltration pool and water intake diagram 1–pool with pumice synthetized guar gum-nanoscale zerovalent iron beads (PPG-nZVI beads) layer; 2–pool without PPG-nZVI beads layer (not shown in the figure); 3–reservoir; 4–water pump; 5–flowmeter; 6–water pipes; 7–sand layer; 8–sawdust layer; 9–pumice layer; 10–fine gravel layer; 11–PPG-nZVI beads layer; 12–partition; 13–pebble support layer; 14–outlet; 15–geotextile.

## 3. Results and Discussion

### 3.1. Sorption Kinetics

The effect of shaking time (0–48 h) on the adsorption of Pb$^{2+}$, Cu$^{2+}$, or Zn$^{2+}$ (initial concentration, 20.0 mg·L$^{-1}$) to PPG-nZVI beads (dosage, 5.0 g·L$^{-1}$) was shown in Figure 2. Equilibrium achieved at about 10 h for Zn$^{2+}$, 6 h for Pb$^{2+}$, and 2.5 h for Cu$^{2+}$. The fitting parameters of kinetic curves are shown in Table 1. For most of the metals (i.e., Cu$^{2+}$ and Pb$^{2+}$), the pseudo-second-order model fits better than pseudo-first-order model model (based on the larger value of $r^2$). For Zn$^{2+}$, there is not much of the difference between two models for describing sorption ($r^2$ value for pseudo-first-order model and pseudo-second-order model is 0.99 and 0.98, respectively). Since pseudo-second-order model is the most commonly used model to describe sorption of heavy metals to heterogenous surfaces, it could better describe sorption of the heavy metals to beads than pseudo-first-order model in our case. Adsorption rate of each heavy metal ion by the PPG-nZVI beads is different, and the $k_2$ decreased follows in the order of Cu$^{2+}$ > Pb$^{2+}$ > Zn$^{2+}$. Azzam et al. suggested that the adsorption mechanism of zero-valent iron towards metal ions is mainly dependent on the standard reduction potential and chemical speciation of the adsorbate under the operating pH [4]. For metal cations such as Zn$^{2+}$/Zn$^0$

($-0.76$ V, 298 K), whose standard potential is negative or close to $Fe^{2+}/Fe^0$ ($-0.44$ V, 298 K). Electrostatic attraction and chemical adsorption, precipitation (with hydroxide ions) and co-precipitation within iron corrosion products are likely the primary mechanisms. While for both reduction potentials of $Cu^{2+}/Cu^0$ ($+0.34$ V, 298 K) and $Pb^{2+}/Pb^0$ ($-0.13$ V, 298 K) are well above of $Fe^{2+}/Fe^0$ ($-0.44$ V, 298 K), and the uptake of $Cu^{2+}$ and $Pb^{2+}$ ions would be expected to primarily take place via a redox mechanism.

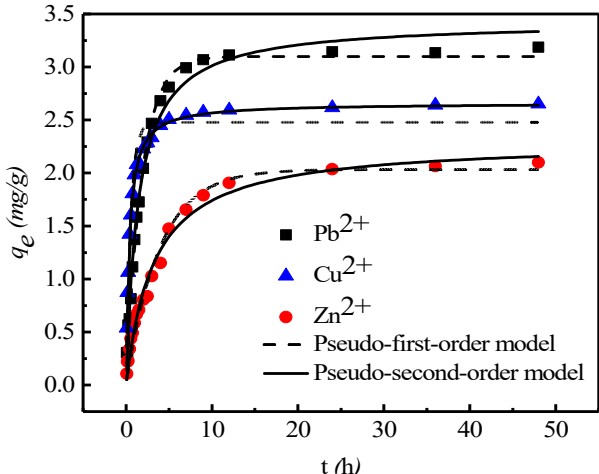

**Figure 2.** Adsorption kinetics of $Pb^{2+}$, $Cu^{2+}$, or $Zn^{2+}$ removal by PPG-nZVI beads. Initial $Pb^{2+}$, $Cu^{2+}$, or $Zn^{2+}$ concentrations, 20 mg·$L^{-1}$; adsorbent dose, 4.0 g·$L^{-1}$; total solution volumes, 400 mL; pH, 5.0; temperature, 25 °C, and shaking time, 48 h.

**Table 1.** Adsorption kinetic constants obtained by different models.

|  | Pseudo-First-Order Model | | | Pseudo-Second-Order Model | | |
|---|---|---|---|---|---|---|
|  | $k_1$ (L/min) | $q_e$ (mg/g) | $r^2$ | $k_2$ (g/mg·min) | $q_e$ (mg/g) | $r^2$ |
| $Pb^{2+}$ | $0.54 \pm 0.01$ | $3.10 \pm 0.02$ | 0.97 | $0.21 \pm 0.01$ | $3.43 \pm 0.06$ | 0.98 |
| $Cu^{2+}$ | $1.83 \pm 0.14$ | $2.47 \pm 0.04$ | 0.94 | $1.03 \pm 0.02$ | $2.66 \pm 0.01$ | 0.99 |
| $Zn^{2+}$ | $0.25 \pm 0.01$ | $2.03 \pm 0.05$ | 0.99 | $0.01 \pm 0.01$ | $2.30 \pm 0.05$ | 0.98 |

*3.2. Sorption Isotherms*

In most of adsorption studies, adsorption models such as Langmuir and Freundlich isotherms have been widely used to evaluate adsorption phenomena [9]. In this work, we use Langmuir and Freundlich model to fit the adsorption data. Sorption isotherms and fitting parameters for the examined metals are shown in Figure 3 and Table 2, respectively. The isotherms of $Pb^{2+}$, $Cu^{2+}$, and $Zn^{2+}$ adsorbed by PPG-nZVI beads are found to better fit with the Langmuir model, judging from the higher correlation coefficients ($r^2 > 0.98$) than Freundlich model (Table 2). The well fit by Langmuir model suggested a monolayer adsorption of above heavy metal ions on the surface of PPG-nZVI beads. The Freundlich equation is based on adsorption on a heterogeneous surface [10] and it is considered to be a multi-layer process. This may be the reason why the Freundlich equation cannot better describe the adsorption of $Pb^{2+}$, $Cu^{2+}$, or $Zn^{2+}$ on PPG-nZVI beads.

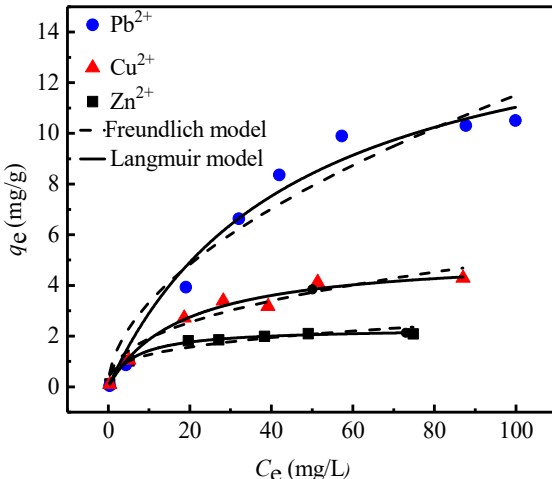

**Figure 3.** Adsorption isotherms of $Pb^{2+}$, $Cu^{2+}$, or $Zn^{2+}$ onto PPG-nZVI beads. Initial $Pb^{2+}$, $Cu^{2+}$, or $Zn^{2+}$ concentrations, 1–80 mg·$L^{-1}$; adsorbent dose, 5 g·$L^{-1}$; total solution volumes 50 mL; pH, 5.0; temperature, 25 °C, and shaking time, 24 h.

**Table 2.** Langmuir and Freundlich isotherm parameters for adsorption of $Pb^{2+}$, $Cu^{2+}$, and $Zn^{2+}$ by PPG-nZVI beads.

| | Langmuir Model | | | Freundlich Model | | |
|---|---|---|---|---|---|---|
| | $b_1$ (L/min) | $q_m$ (mg·$g^-$) | $r^2$ | $K_f$ (mg$^{1-1/n}$·$L^{1/n}$·$g^{-1}$) | $n$ (mg·$L^{-1}$) | $r^2$ |
| $Pb^{2+}$ | 0.02 ± 0.01 | 16.01 ± 1.84 | 0.98 | 0.72 ± 0.21 | 2.39 ± 0.43 | 0.93 |
| $Cu^{2+}$ | 0.05 ± 0.01 | 5.23 ± 0.40 | 0.98 | 0.61 ± 0.15 | 3.23 ± 0.70 | 0.89 |
| $Zn^{2+}$ | 0.16 ± 0.01 | 2.31 ± 0.04 | 0.99 | 0.96 ± 0.36 | 1.85 ± 0.31 | 0.93 |

As depicted, there are pronounced differences for sorption capacities of the PPG-nZVI bead to the different metals examined (Figure 3). The maximum adsorption capacities ($q_m$) for $Pb^{2+}$, $Cu^{2+}$ and $Zn^{2+}$ obtained by Langmuir isotherm were 16.01, 5.23, 2.31 mg/g, respectively (Table 2). In the above section, we observed that the adsorption rates decreased in the order of $Cu^{2+}$ > $Pb^{2+}$ > $Zn^{2+}$, in consistent with the order of the standard reduction potential. However, for the adsorption capacity, the $q_m$ decreased following in the order of $Pb^{2+}$ > $Cu^{2+}$ > $Zn^{2+}$, not well correlated with the order of standard reduction potential. Besides the influence of reduction potential, other factors such as the electronegativity, paramagnetic, and atomic weight could all make contribute to affect their sorption capacity. The maximum adsorption capacity for $Pb^{2+}$ was much higher than the values of $Cu^{2+}$ and $Zn^{2+}$, showing the following capacity order: $Pb^{2+}$ > $Cu^{2+}$ > $Zn^{2+}$. $Pb^{2+}$ has the preferential uptake, which may be because of the higher affinity of PPG-nZVI beads towards $Pb^{2+}$ than the other two heavy metal ions. A previous study by Wang et al. [11]. showed that the sorption of $Pb^{2+}$, $Cu^{2+}$ and $Cd^{2+}$ by gelation with alginate following the same order of $Pb^{2+}$ > $Cu^{2+}$ > $Cd^{2+}$. They explained the difference is due to the strength of covalent bonds between the metal ions and beads that follows the sequence of $Pb^{2+}$ > $Cu^{2+}$. The highest adsorption capacity of the PPG-nZVI beads for $Pb^{2+}$ may also because of the more electronegative ion (2.33), paramagnetic and the larger atomic weight (207.2) and the higher standard reduction potential of $Pb^{2+}$ (−0.13 V) compared to $Zn^{2+}$ (−0.76 V). These properties make $Pb^{2+}$ most likely to be adsorbed. The PPG-nZVI beads had the least affinity for $Zn^{2+}$ may be because of its lowest electronegativity [$Pb^{2+}$ (2.33) > $Cu^{2+}$ (1.90) > $Zn^{2+}$ (1.65)] and largest hydrated ionic radii [$Pb^{2+}$ (0.401 nm) < $Cu^{2+}$ (0.419 nm) < $Zn^{2+}$ (0.430 nm)]. The affinity order ($Pb^{2+}$ > $Cu^{2+}$ > $Zn^{2+}$) by PPG-nZVI beads was in accordance with the reverse order of hydrated ionic radii and Pauling's electronegativity. Similar results were found by Wang et al. who studied the adsorption of $Pb^{2+}$, $Cu^{2+}$, and $Cd^{2+}$ ions by wheat-residue-derived black carbon and found the

affinity order of $Pb^{2+} > Cu^{2+} > Cd^{2+}$. They explained the phenomenon by the hard soft acids and bases (HSAB) principle [12]. Choy and McKay studied the sorption of metal ions onto bone char, and found the selectivity of bone char followed in the order of $Cu^{2+} > Cd^{2+} > Zn^{2+}$ [13]. Yu et al. studied the sorption of metal ions onto modified sugarcane bagasse, and their result was also matched with the reverse order of the hydrated ionic radii: $Pb^{2+} > Cu^{2+} > Cd^{2+} > Zn^{2+}$ [14].

### 3.3. Effect of pH and Ionic Strength on Sorption

Aqueous phase pH would govern the adsorption process as it affects the surface charge of the adsorbent, the degree of ionization and the dominant speciation of the metal contaminant [4]. Besides, a change of pH could influence the reaction rate of iron oxidation [15]. Thus, the effect of pH on heavy metals removal by PPG-nZVI beads was conducted over a pH range of 3.0–5.5. For pH above 5.5, adsorption experiments were not conducted because the precipitation for $Cu^{2+}$ could occur. As seen from Figure 4a, the removal efficiency of $Pb^{2+}$, $Cu^{2+}$, and $Zn^{2+}$ by PPG-nZVI beads increased with the increase of pH. From pH 3.0–5.5, the removal rate increased from 49.0–92.7%, 21.7–81.9%, and 18.6–48.4% for $Pb^{2+}$, $Cu^{2+}$, and $Zn^{2+}$, respectively. The results indicated that the removal of different heavy metal ions by PPG-nZVI beads was strongly pH-dependent. In $Fe^0$ treatment systems, the removal mechanisms of $Pb^{2+}$, $Cu^{2+}$, and $Zn^{2+}$ are generally believed to involve the adsorption of the ions on the iron surface where electron transfer takes place. $Pb^{2+}$, $Cu^{2+}$, or $Zn^{2+}$ is reduced to the formation of $Pb^0$, $Cu_2O$, $Cu^0$, or $Zn^0$ with the oxidation of $Fe^0$ to $Fe^{3+}$ or $Fe^{2+}$ under acidic conditions [16,17]. Actually, besides the removal mechanisms regarding to the electron transfer between heavy metals and iron surface, the heavy metal might also interact with PVA molecules and guar gum by electrostatic interaction followed by bridging mechanisms. The explanation is similar with Lapointe and Barbeau [18]. Divalent cations such as $Pb^{2+}$, $Cu^{2+}$, and $Zn^{2+}$ is expected to have an important impact during the aggregation of negatively charged beads with anionic polyelectrolytes. It is likely that such metal ions will act as binding agent between anionic polyelectrolyte sites and negative particles (i.e., complexing of metal ions with anionic carboxyl groups), despite the anticipated electrostatic repulsion. The interactions may refer to electrostatic interaction, hydrogen bonding, and/or divalent cation binding. PVA is composed of copolymers of vinyl alcohol and vinyl acetate. The high density of hydroxyl groups on its structure can lead to a more cross-linked configuration. The interparticle bridging behavior in the presence of nonionic PVA is essentially due to hydrogen bonding. The observed lower uptake in the acidic medium may be attributed to the partial protonation of the active groups and the competition of $H^+$ with metal ions for adsorption sites on the PPG-nZVI beads. The increase of pH results in an increase the amount of hydroxyl groups. The number of negatively charged sites on PPG-nZVI beads was improved, leading to the enhanced attraction force between metallic cations and the beads. As a result, the removal efficiency of $Pb^{2+}$, $Cu^{2+}$ and $Zn^{2+}$ by beads increased. On the other hand, metals are immobilized via precipitation by hydrolyzing as metal hydroxides at high pH [4].

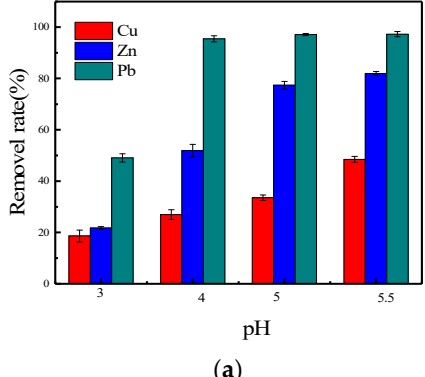
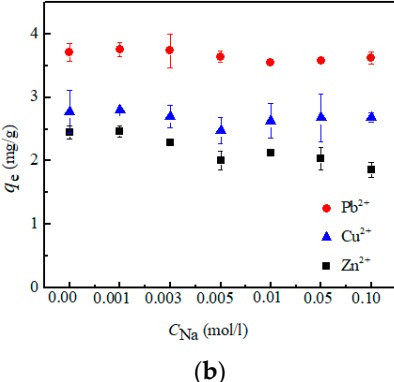

(**a**)         (**b**)

**Figure 4.** Effect of pH (**a**) and ionic strength (**b**) on removal of $Pb^{2+}$, $Cu^{2+}$, or $Zn^{2+}$ by PPG-nZVI beads.

From Figure 4b, it is evident that the removal efficiency of $Pb^{2+}$, $Cu^{2+}$, or $Zn^{2+}$ by PPG-nZVI beads was not obviously affected by the presence of ionic strength from 0–0.1 M in $NaNO_3$ background solution. The result is in accordance with the previous study which indicated that the adsorption of $Pb^{2+}$, $Cu^{2+}$, or $Zn^{2+}$ onto PPG-nZVI beads may be governed by inner-sphere complex adsorption mechanism. In detail, as the concentration of $NaNO_3$ raised from 0 to 0.003 M, the adsorption capacity of $Pb^{2+}$ or $Cu^{2+}$ was not affected. However, for $Zn^{2+}$, sorption capacity decreased from 2.44 to 2.29 mg·g$^{-1}$. The depression of $Zn^{2+}$ adsorbed by PPG-nZVI beads might be due to the competition of $Na^+$ with $Zn^{2+}$ for the same binding sites on the PPG-nZVI beads surface, which further confirmed that adsorption is main process to remove $Zn^{2+}$ but not that important for $Pb^{2+}$ and $Cu^{2+}$. When the concentration of $NaNO_3$ raised from 0.003 to 0.005 M, the adsorption capacity of $Pb^{2+}$ and $Cu^{2+}$ started to decrease, while it continually decreased for $Zn^{2+}$. It indicated the competition occurs on $Pb^{2+}$, $Cu^{2+}$, or $Zn^{2+}$. As literature reported, the much influenced by ionic strength on the uptake of heavy metal ions could be explained by the formation of iron oxide. With the concentration of $NaNO_3$ raised from 0.005 to 0.1 M, the uptake capacity increased for $Cu^{2+}$ and $Pb^{2+}$ which indicated that the electrostatic double layer of the PPG-nZVI beads is more compressed leading to the more adsorption. Based on the above analysis, it is concluded that the adsorption of $Pb^{2+}$, $Cu^{2+}$ by PPG-nZVI beads proceeds through both outer-sphere adsorption and co-precipitation and inner-sphere reduction. Adsorption and co-precipitation might play an important role on the interaction between $Zn^{2+}$ and the beads.

### 3.4. Characterization before and after Sorption

FTIR spectra analysis on PPG-nZVI beads was conducted to identify the main functional groups involved in removal of the $Pb^{2+}$, $Cu^{2+}$ or $Zn^{2+}$ ions (Figure 5). The fresh PPG-nZVI beads were used as control samples to compare the shifts of FTIR peaks after reaction for identification of participated functional groups in metal uptake. The main PPG-nZVI beads bands observed in the FTIR spectra (a) were in accordance with previous report [7]. As shown in Figure 5, the band at 3436 cm$^{-1}$ is the characteristic peak of –OH stretching vibrations and functional group –OH usually forms a broad band region at the wavenumber of 3100–3700 cm$^{-1}$, which is attributed to the adsorption of water on nZVI and the exist of alcoholic and carboxyl groups [4,7]. For PPG-nZVI beads after adsorption of the metal ions, the peak at 3436 cm$^{-1}$ was shifted to 3431 cm$^{-1}$, 3430 cm$^{-1}$, and 3427 cm$^{-1}$ for adsorption of $Pb^{2+}$, $Cu^{2+}$, and $Zn^{2+}$, respectively and the wave number decreased. This may be ascribed to the combination between the metal ions and the –OH group [19]. The vibrational band observed at 2919 cm$^{-1}$ referred to the stretching –CH from alkyl groups. The peak at wavelength of 1715 cm$^{-1}$ was due to the stretching vibration of carbonyl double bond. Two adsorption peaks at a wavenumber of 1713 cm$^{-1}$ observed in the PPG-nZVI beads after adsorption of $Cu^{2+}$ and $Pb^{2+}$ may be due to the stretching vibration of carbonyl double bond (–C=O) and carbon oxygen single bond (–C–O) in carboxyl groups, respectively. These results indicated that the alcoholic, carboxyl, alkyl groups in PVA and carboxylate ions in guar gum might be involved in the metal binding. While the band at 1635 cm$^{-1}$ was the asymmetrical and symmetrical stretching vibration of COO−. These wave numbers were weakened suggesting the important role of COO− in the metal adsorption.

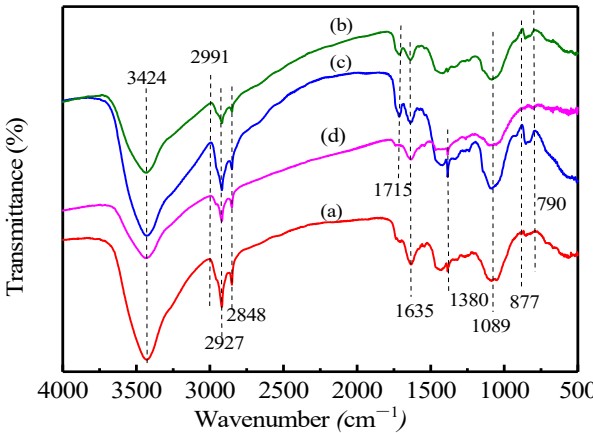

**Figure 5.** FTIR of the PPG-nZVI beads. (**a**) is the result of the PPG-nZVI beads before adsorption. (**b**), (**c**), and (**d**) is the result of the PPG-nZVI beads after adsorption of $Pb^{2+}$, $Cu^{2+}$, and $Zn^{2+}$ respectively. The initial concentration of each heavy metal ion is 50 mg·$L^{-1}$; the dosage of PPG-nZVI beads, 5.0 g·$L^{-1}$; pH, 6.0 and temperature, 25 °C.

Figure 6 shows the peaks of Fe 2p3/2 and Fe 2p1/2 located at 711 and 724 eV for $Fe_2O_3$ and FeOOH [20]. In addition, the satellite peak at 719.81 eV is the characteristic of $Fe_2O_3$. Furthermore, a feature peak of $Fe^0$ can be found at around 707.2 eV, suggesting that $Fe^0$ does exist on the PPG-nZVI beads [21]. FeOOH was a result of surface hydroxylation and exposed shell dehydration when iron nanoparticles contacted with aqueous solution. After the reaction, the peak of $Fe^0$ could not be found for $Pb^{2+}$, $Zn^{3+}$, $Cu^{2+}$, and the peak intensity for $Fe^{3+}$ and $Fe^{2+}$ increased, indicating that $Fe^0$ had been oxidized on the surface of PPG-nZVI beads.

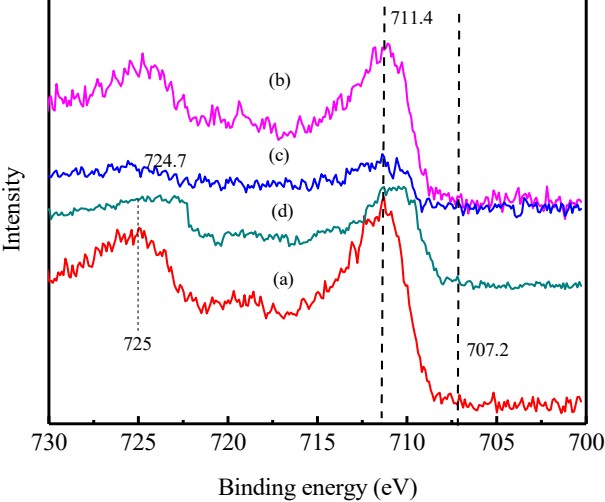

**Figure 6.** XPS spectroscopy of Fe 2p for fresh PPG-nZVI beads (**a**) and for PPG-nZVI beads reaction with $Pb^{2+}$ (**b**), $Cu^{2+}$ (**c**), and $Zn^{2+}$ (**d**).

Figure 7 and Figure S1 show the XPS spectra of the PPG-nZVI beads before and after adsorption of $Cu^{2+}$, $Zn^{2+}$, and $Pb^{2+}$. From Figure 7, the photoelectron peak for the oxygen 1s region resolved into three curves with peaks of approximately 530, 531, and 532 eV, which represented for the binding energies of oxygen in $O_2^-$, $OH^-$ and chemically or physically adsorbed water. That is in accordance with literature. In Figure 7b–d, there was a shift of peak positions after the PPG-nZVI beads reacted with $Cu^{2+}$, $Zn^{2+}$ and $Pb^{2+}$ compared of PPG-nZVI beads before adsorption. The heavy metals sorption was accompanied by the change in oxygen binding, providing evidence that the oxygen-containing functional groups on the surface of the PPG-nZVI beads taking part in the sorption of heavy metals.

After adsorption, the peak at 530.5 moves to 531.4, 530.9, and 531.9 for $Pb^{2+}$, $Cu^{2+}$, and $Zn^{2+}$, respectively. It indicated the exist of $Fe(OH)_3$, $FeOOH$, and $Pb(OH)$ [5].

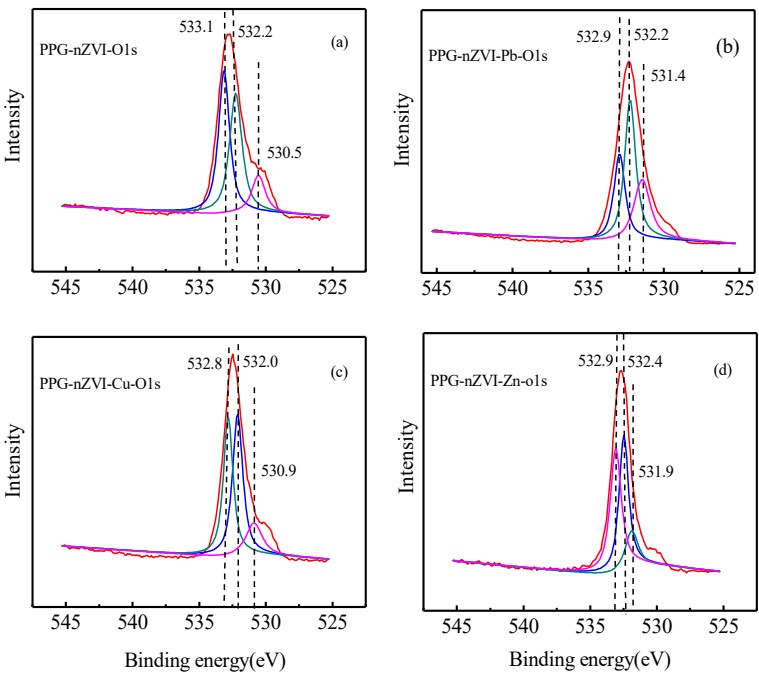

**Figure 7.** XPS spectroscopy of O 1 s for fresh PPG-nZVI beads (**a**) and for PPG-nZVI beads reaction with $Pb^{2+}$ (**b**), $Cu^{2+}$ (**c**), and $Zn^{2+}$ (**d**).

### 3.5. Efficient Removal of Heavy Metals in Stormwater Infiltration Facility

To test the removal of heavy metals by PPG-nZVI beads in remediation system, PPG-nZVI beads as filling materials were paved in the simulated stormwater infiltration pool to examine their removal efficiency on treating the heavy metal solutions including $Pb^{2+}$, $Cr^{6+}$, $Zn^{2+}$, $Cd^{2+}$, and $Cu^{2+}$. Generally, the treatment efficiency of heavy metals by pool 1 with PPG-nZVI beads layer is superior to pool 2 without beads layer, and the removal efficiency decreased following in the order of $Pb^{2+}$ > $Zn^{2+}$ > $Cr^{6+}$ > $Cu^{2+}$ > $Cd^{2+}$ after 3 h operation. As seen from Figure 8, the removal efficiency of $Pb^{2+}$ in pool 1 and pool 2 was not significantly different, and the removal rate of effluent of both pools was more than 84%. After operation of the facility, the removal rate of $Pb^{2+}$ increased slowly. The reason may be due to that $Pb^{2+}$ is more likely to adhere to the surface of the suspended particles than other ions, so that it is easily removed by the filters other than PPG-nZVI beads layer. The removal efficiency of $Zn^{2+}$ was significant for both plant pools. The removal rate of $Zn^{2+}$ increased from 50%–73% to 69%–78% after adding of PPG-nZVI bead layer. There is also a significant difference between pool 1 and pool 2 in the removal of $Cr^{6+}$. The removal efficiency towards $Cr^{6+}$ was significant for both plant pools. The removal rate of $Cr^{6+}$ by pool 1 is above 75%, while the removal rate by pool 2 without PPG-nZVI beads layers is between 60%–65%. It seems that the removal of $Cr^{6+}$ by traditional layers retention such as sand, pumice, and gravel is limited. The addition of PPG-nZVI bead layer could significantly improve the removal rate and effectively reduce the risk of pollution. For $Cd^{2+}$, the pool 1 is slightly superior to pool 2. The removal rate of $Cd^{2+}$ was between 58% and 68% for both pools. The removal behavior of the pool 1 and pool 2 on $Cu^{2+}$ is quite different. The removal rate of $Cu^{2+}$ was 2.3 times higher by pool 1 with beads than pool 2 after 3 h operation. Above all, the pool with PPG-nZVI bead layer can effectively handle high-intensity, high-concentration instantaneous water and indicated that the PPG-nZVI beads might be effective sorbents as the filling materials in stormwater infiltration facility in purification of heavy metal contaminated water. The preparation of PPG-nZVI beads requires only chemicals including binder and porous material for the ball formation. The chemicals were all environmental

friendly thus could reduce pollution. The cost of the synthesized sorbents is estimated to be only 0.566 $ per 1000 beads, which could treat 40 mg of heavy metal pollutants in water system. Compared with the nZVI powder, PPG-nZVI beads could greatly improve the effective contact time with the pollutant. Especially when they were applied as the filling materials in stormwater filtration devices, the well-formed beads could allow the polluted water flowing through the filler with good penetration.

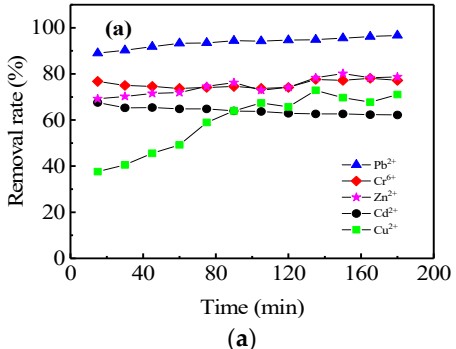 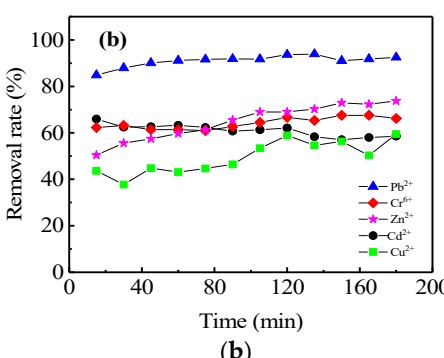

(**a**)  (**b**)

**Figure 8.** Effect of simulated stormwater infiltration pool on heavy metal removal. (**a**) pool 1 (with the PPG-nZVI bead filter layer); (**b**) pool 2 (without the PPG-nZVI bead filter layer).

## 4. Conclusions

The large adsorption capacity and easily solid-liquid separation property of PPG-nZVI beads make them superior sorbents for practical application in water treatment devices such as stormwater infiltration facility. The adsorption kinetics data could be well fitted by the pseudo-second-order model for all kinds of heavy metals and the adsorption rate decreased followed in the order of $Cu^{2+} > Pb^{2+} > Zn^{2+}$, which is consistent with their reduction potential. Langmuir isotherm fits the data well and the maximum adsorption capacities decreased followed in the order of $Pb^{2+} > Cu^{2+} > Zn^{2+}$ (i.e., 16.02, 5.24, and 2.31 $mg \cdot g^{-1}$) with the bead dosage of 5 $g \cdot L^{-1}$, respectively. The pH was found to have significant effect on the removal process. The maximum sorption capacity was observed at pH 5.5, indicating that competition occurred between $H^+$ and metal ions. While the ionic strength did not significantly influence the sorption process. The simulated stormwater infiltration pool with PPG-nZVI beads layer show large removal efficiency on treating the contaminated water containing $Pb^{2+}$, $Cu^{2+}$, $Zn^{2+}$, $Cr^{6+}$ and $Cd^{2+}$ and the removal rates were more than 65% at least. The results indicated that the PPG-nZVI beads might be effective sorbents as the filling materials in stormwater infiltration facility in purification of heavy metal contaminated water.

**Supplementary Materials:** The following are available online at http://www.mdpi.com/2076-3417/9/20/4213/s1, Figure S1: Typical wide scan XPS spectra for fresh PPG-nZVI beads (a) and for PPG-nZVI beads reaction with $Pb^{2+}$ (b), $Cu^{2+}$ (c) and $Zn^{2+}$ (d).

**Author Contributions:** Conceptualization, X.Z.; methodology, X.Z. and L.Y. software, L.Y., X.Z., and J.L.; validation, L.Y.; formal analysis, X.Z. and L.Y.; writing, L.Y. and X.Z.; review, J.L., Z.Z., and C.T.

**Funding:** This research was funded by the Construction of High Level Teaching Teams in Universities of Beijing-the Youth Top-Notch Talent Cultivation Program (CIT&TCD201804051); National Natural Science Foundation of the China-Youth Project (51508017 and 51708014); the Fundamental Research Funds for Beijing University of Civil Engineering and Architecture (ZF14042 and X19036); the Fundamental Research Funds of Beijing Vocational College of Agriculture (XY-XF-17-13); Pyramid Talent Cultivation Project of Beijing University of Civil Engineering and Architecture.

**Conflicts of Interest:** The authors have declared no conflict of interest.

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
