# Peer review of "Removal of Different Kinds of Heavy Metals by Novel PPG-nZVI Beads and Their Application in Simulated Stormwater Infiltration Facility"

_applsci, doi:10.3390/app9204213_

Round 1

Reviewer 1 Report

This manuscript presents a good idea of assessment of (selected) heavy metals removal from stormwater using new reactive material. The work includes a valuable data that broaden the current knowledge in the field of application of synthesized materials. I believe that the manuscript can be considered for publication in this journal, however some minor revisions (specified below) have to be made:

"2.1 Materials" section: In this section, the pH value of initial solutions used in kinetic and equilibrium tests should be given. Section 2.4: Usually, when we talk about the size of grains we use phrase "particle size" or "grain size". Authors used the term "diameter" which is a little bit confusing. Section 3.1, lines 143-144: the Authors claimed that the PSO model fits better that PFO moder (based on the value of r2).  Assuming this assessment criterion, this is not true for Zn. Section 3.1, line 144: Could you explain what do you mean by "adsorption rates of each heavy metal"? Do you mean "qe"? If yes, the order is not correct (this order is correct only for k2). Figure 2: the maximum shaking time was 48 h. For most of the samples the shaking time was shorter. Section 3.2: the first sentence is not clear. Section 3.2, lines 167-168: why the Freundlich equation could not better describe the sorption of heavy metals? It is not clear from this sentence. All Figures are difficult to read. They should be bigger. Also the descriptions in the figures should be written with larger font. Line 186: Authors explained the higher adsorption capacity for Pb than for Zn through the higher standard reduction potential of Pb compared to Zn. The value of this potential for Cu is even higher than for Pb and the adsorption capacity is lower for Cu than for Zn. Could you explain why, in your opinion, the relation that you mentioned about (relation between adsorption capacity and standard reduction potential) doesn't work for Cu? Could you explain why did you use NaNO3 in tests of ionic strength effect on adsorption? In stormwater treatment systems, the ionic strength is usually caused by NaCl.  Authors consider the application of PPG-nZVI in infiltration systems for stormwater treatment. However, that kind of application usually requires a large volume of sorbents. Have you analyzed the costs of PPG-nZVI synthesis?  

Author Response

Dear reviewer, the reply was in the attached word file. Please see the attachment. Thank you.

Reviewer 2 Report

General comment:

PPG-nZVI beads were synthesized and tested for the removal of metal ions. The impact of pH and ionic strength were also covered. PPG-nZVI beads is used as a new adsorbent for polluted stormwater runoff. The removal is quite promising: > 65% for Pb2+, Cu2+, Zn2+, Cr6+ 23 and Cd2+. This paper should definitely be considered for publication in Applied Sciences. The following recommendations might improve the manuscript. Extra explanations must be added about the adsorption mechanisms e.g., specific interactions with functional groups at the polyvinyl alcohol/guar gum nanobeads surface. 

Specific comments:

Reference is required for lines 30-33

Section 2.1: PVA is added to the structure by precipitation. Is this material is sensitive to shearing? The authors could comment more deeply on the material mechanical resistance.

Except the interaction with iron via electron transfer (mentioned in line 209), Pb, Cu, Zn and Cr (hydr)oxides might interact with PVA (at C=0 bonds) and/or with polysaccharides (guar gum). Clear comments are needed on this. Please see the different adsorption/aggregation pathways here:

https://www.sciencedirect.com/science/article/pii/S1383586619319604

Tables 1 and 2

Author Response

(The authors gave the same response as above.)

Reviewer 3 Report

A study focusing a heavy metal removal by PPG-nZVI is proposed. The topic is interesting and the study is of good quality. I suggest enlarging the state-of-the-art about the topic and improve the quality of figures. Also English deserves a careful check. Other minor suggestions in the attached file.

Author Response

(The authors gave the same response as above.)
